# Nausea and Vomiting during Early Pregnancy among Chinese Women and Its Association with Nutritional Intakes

**DOI:** 10.3390/nu15040933

**Published:** 2023-02-13

**Authors:** Shujing Zhu, Ai Zhao, Hanglian Lan, Pin Li, Shuai Mao, Ignatius Man-Yau Szeto, Hua Jiang, Yumei Zhang

**Affiliations:** 1Department of Nutrition and Food Hygiene, School of Public Health, Peking University, Beijing 100191, China; 2Vanke School of Public Health, Tsinghua University, Beijing 100091, China; 3Yili Innovation Center, Inner Mongolia Yili Industrial Group Co., Ltd., Hohhot 010110, China; 4School of Nursing, Peking University, Beijing 100191, China; 5Beijing Key Laboratory of Toxicological Research and Risk Assessment for Food Safety, School of Public Health, Peking University, Beijing 100191, China

**Keywords:** morning sickness, nutritional intake, pregnancy trimester, first, gestational weight gain

## Abstract

Nausea and vomiting in pregnancy (NVP) is one of the most common uncomfortable symptoms of women in early pregnancy. A total of 303 Chinese pregnant women from 10 urban cities in their first trimester were recruited in this study to collect their sociodemographic characteristics and their NVP occurrence. Their dietary nutrient and food intakes were also collected by a 24 h dietary recall and a semi-quantitative food frequency questionnaire (SFFQ). Using the univariate analysis and multiple linear regression analysis to estimate the correlation between NVP and dietary intake, we found that 255 (84.1%) pregnant women experienced NVP during their first trimester. The intake of energy, protein, fat, vitamin A, thiamin, riboflavin, vitamin E, phosphorus, potassium, iron and zinc was lower in women with NVP than in those with no NVP. Additionally, women with NVP were more likely to have insufficient intake of protein, riboflavin, calcium, phosphorus and selenium. In terms of specific food groups, the average daily intake of mushrooms, algae, nuts and seeds, meat, eggs and dairy products in the NVP group was lower. Women in the severe NVP group even had insufficient gestational weight gain. We should pay more attention to women who experience nausea and vomiting during pregnancy and provide them with targeted nutritional support.

## 1. Introduction

Nausea and vomiting in pregnancy (NVP) is one of the uncomfortable symptoms of women in early pregnancy. About 35–91% of pregnant women in the world will experience varying degrees of nausea or vomiting during pregnancy, generally starting at 6–8 weeks of pregnancy and ending at 16–20 weeks of pregnancy [1,2]. Moderate and severe nausea or vomiting, also known as hyperemesis gravidarum (HG), not only causes a decrease in the appetite of pregnant women but also causes electrolyte disorder, acid–base imbalance and even weight loss [3], reducing the quality of sleep and life, even impairing daily function [4,5]. In addition, NVP, especially HG, can cause adverse pregnancy outcomes such as preterm birth, small-for-gestational-age infants and an increased risk of neurodevelopmental delay [6], which may be due to a lack of key nutrients for the growth and development of mothers and infants [7].

There are relatively few studies on the dietary intake of pregnant women with NVP, and most of these studies focus on HG [6]. However, the incidence of HG is only 0.8–3.3%. Based on this fact, focusing only on HG may actually ignore a large proportion of pregnant women. Current studies have also shown inconsistent results for changes in dietary intake caused by NVP. A Norwegian study of more than 50,000 pregnant women found that women with NVP had higher intakes of energy, carbohydrates, vitamin A, folate and vitamin C [8]. However, studies in other countries, such as South Korea [9], Finland [10] and the United Kingdom [11], found that the intake of energy, macronutrients and micronutrients, as well as the diet quality, were significantly lower in women with NVP women than in those with no NVP. The inconsistency of research results may be due to cross-sectional data or different regional cultures. Moreover, there is no analysis of the association between NVP and dietary intake in China. Therefore, it is necessary to explore the association between NVP and dietary intake in Chinese pregnant women.

In conclusion, it is not clear which nutrients are key to women with NVP, the degree of nutrient deficiencies caused by NVP, the specific food groups that cause nutrient deficiencies or the nutrient intake of pregnant women with different NVP severity. Therefore, more research on the dietary intake of women with NVP is needed.

The purpose of our study was to analyze the incidence of NVP among pregnant women in the first trimester in 10 Chinese cities and the association between NVP, dietary intake and gestational weight gain.

## 2. Materials and Methods

### 2.1. Subjects

This study was based on a cross-sectional survey, the “Survey on Nutritional Status of Pregnant Nursing Mothers and Infants Aged 0–36 Months in 10 Cities in China”, which was conducted from 2019 to 2020. According to the geographical location and economic development level, two first-tier cities (Beijing and Guangzhou), three new first-tier cities (Suzhou, Chengdu and Shenyang), three second-tier cities (Ningbo, Lanzhou and Nanchang), one third-tier city (Hohhot) and one fourth-tier city (Xuchang) were selected. In each city, one hospital or one maternal and child health care center was selected, and each city planned to recruit 30 pregnant women in the first trimester. The inclusion criteria were healthy pregnant women aged 20–45 years who were non-alcoholic and non-smokers. The exclusion criteria were pregnant women with gestational diabetes mellitus, hypertension, infectious diseases, mental illness and memory impairment, inability to answer questions and who were taking medication (excluding nutritional supplements) during the survey period. Finally, 307 pregnant women in the first trimester were included. After excluding 4 participants with missing data on nausea or vomiting during pregnancy, our analysis was based on 303 pregnant women in the first trimester. Based on the calculation formula of sample size for cross-sectional study:(1)$$n=\frac{{Z^{2}}_{1-\alpha /2}\times p(1-p)}{d^{2}}$$
where *n* was sample size required, *α* was significant level set at 0.05, *p* was estimated prevalence of nausea and vomiting in pregnancy set as 0.718 according to a previous study conducted in China [12] and *d* (allowable error) was 0.1 × *p*. Here, the minimum theoretical sample size was calculated as around 151. Finally, a total of 303 individuals were included, satisfying the calculated sample size requirement.

### 2.2. Data Collection

The survey was conducted through a face-to-face questionnaire survey, and all of the questionnaire investigators received unified training.

#### 2.2.1. Nausea or Vomiting during Pregnancy

With the assistance of the investigators, pregnant women were divided into an NVP group and a non-NVP group by the question “Have you experienced nausea or vomiting since the beginning of pregnancy?” If the answer was “Yes”, then the woman was assigned to the NVP group. If the answer was “No”, then the woman would be assigned to the non-NVP group. Moreover, we asked the woman who answered “Yes” an additional question, “Is your nausea or vomiting mild, moderate or severe?” Based on the self-evaluation of the severity, the NVP group was subdivided into the mild, moderate, and severe NVP groups.

#### 2.2.2. Dietary Data

With the aid of a standard food reference map, standard bowls and plates [13], a one-time 24 h dietary recall was used to record the types and quantities of all foods, beverages and condiments consumed by the participants over the previous 24 h. The intake of nutrients was calculated based on the 24 h dietary recall according to the Chinese food composition table [14].

In addition, we used a 36-item semi-quantitative food frequency questionnaire (SFFQ) to evaluate participants’ dietary food intake over the past month. All food items were categorized into 15 food groups: cereals; potatoes; fresh vegetables; pickled and fermented vegetables; mushroom and algae; soybeans and their products; nuts and seeds; fruits; livestock meat; poultry meat; aquatic products; eggs; dairy products; meat and fish bone soup; and snacks and beverages according to the recommended health industry standard in China, “Specification for expression of food composition data” [15] (Appendix A). The daily intakes for each food group were then calculated.

#### 2.2.3. Demographic Data

Essential characteristics of the pregnant women were investigated by questionnaires, including their age, education, family monthly per capita income, city of residence, gestational week, gravidity, parity, etc. The trained investigators used identical equipment, such as height-measuring instruments and weight-measuring instruments, to measure and record the height and weight of pregnant women. We asked our subjects to recall their weight before pregnancy in order to calculate the pre-pregnancy body mass index (BMI), gestational weight gain (GWG) and the rate of GWG per week. According to the Chinese BMI standards, underweight, normal weight and overweight/obesity were defined as <18.5 kg/m^2^, 18.5–23.9 kg/m^2^ and ≥24 kg/m^2^, respectively.

### 2.3. Statistics Analysis

SPSS version 27.0 was used for statistical analysis. Values for categorical variables were presented as N (%). According to the normality test results of continuous data, values were presented as the mean ± standard deviation (SD) or median (P25, P75). The chi-square test was used to compare the characteristics of women in the different NVP groups, and the Mann–Whitney U test was used to compare the dietary intake and GWG of the women in the different groups. We used Spearman’s rank correlation analysis to explore the trend of the effect of NVP severity on dietary intake and GWG. Multiple linear regression models were used to obtain the correlation between NVP and maternal dietary intake, adjusted by the relevant demographic characteristics (age, education, income, city, pre-pregnancy BMI, gravidity, parity). The statistically significant difference in this study was set at 0.05.

## 3. Results

### 3.1. Incidence of Nausea and Vomiting in Women during Early Pregnancy

A total of 303 pregnant women in the first trimester were included in this study, of whom 255 were in the NVP group who had experienced nausea or vomiting since the beginning of pregnancy, accounting for 84.1% of the study population, divided into mild (62.4%), moderate (25.9%) and severe (11.8%) subgroups according to the self-evaluation of pregnant women. Pregnant women in first-tier cities were more likely to experience NVP (Table 1).

### 3.2. The Association between NVP and Dietary Nutrient Intake

Compared with those in the non-NVP group, pregnant women in the NVP group generally had a lower dietary nutrient intake, as reflected in the lower intake of energy, protein, fat, vitamin A, thiamin, riboflavin, vitamin E, phosphorus, potassium, iron and zinc (Table 2).

After adjustment for demographic characteristics, including age, education, income, city, pre-pregnancy BMI, gravidity and parity, the NVP group had a reduction in energy of 299.3 (87.7–510.8) kcal/day and a reduction in protein of 18.6 (8.1–29.1) g/day compared with the non-NVP group. Other nutrients, such as fat, thiamine, riboflavin, vitamin E, calcium, phosphorus, potassium, iron, zinc and selenium, also had a decrease (Table 3). The complete linear regression models are shown in Appendix A.

The effect of NVP on dietary intake was also different with different severity. Compared with pregnant women in the non-NVP group, the intake of fat, thiamin, riboflavin, vitamin E, phosphorus and potassium was lower in the mild NVP group; the intake of protein, fat, cholesterol, thiamin and zinc was lower in the moderate NVP group; and the intake of protein, fat, carbohydrate, dietary fiber, vitamin A, thiamin, riboflavin, vitamin E, calcium, phosphorus, potassium, iron, zinc, selenium and manganese was lower in the severe NVP group. The results of Spearman’s correlation analysis showed that the more severe the nausea/vomiting in pregnant women in early pregnancy, the lower the intake of energy, protein, fat, carbohydrate, vitamin A, thiamin, riboflavin, niacin, vitamin E, calcium, phosphorus, potassium, iron, zinc, selenium and manganese (Appendix A). After adjustment for demographic characteristics, we also found a similar result that women in the severe NVP group tended to have the lowest nutrient intake (Appendix A).

Comparing the nutrients’ adequacy rate between pregnant women with and without nausea and vomiting in early pregnancy, we found that among pregnant women with NVP, the intake adequacy rate of all nutrients was lower than that of pregnant women without NVP. However, statistically significant differences were only found in riboflavin, calcium, phosphorus and selenium (Table 4). After controlling for covariates, we found that NVP was linked to a lower likelihood of adequate intake of protein (OR = 0.45, 95% CI 0.22 to 0.90, *p* = 0.023), riboflavin (OR = 0.34, 95% CI 0.14 to 0.83, *p* = 0.018), calcium (OR = 0.30, 95% CI 0.12 to 0.71, *p* = 0.007), phosphorus (OR = 0.32, 95% CI 0.15 to 0.69, *p* = 0.004) and selenium (OR = 0.39, 95% CI 0.16 to 0.96, *p* = 0.041).

### 3.3. The Association between NVP and Dietary Food Intake

Compared with pregnant women in the non-NVP group, the daily intake of mushrooms, algae, nuts and seeds, meat, eggs and dairy products in the NVP group was lower (Table 5). The effects of the different severities of NVP on dietary food intake are also different. Compared with the non-NVP group, the intake of nuts, seeds, livestock meat and eggs was lower in the mild NVP group; the intake of mushroom, algae and poultry meat was lower in the moderate NVP group; and the intake of fresh vegetables, meat, poultry, eggs and dairy products was lower in the severe NVP group. The results of Spearman’s correlation analysis showed that the more severe the nausea/vomiting in the first trimester of pregnancy, the lower the intake of nuts and seeds, livestock meat, poultry meat, aquatic products, eggs and dairy products (Appendix A). After adjusting the demographic characteristics, a similar result showed that compared with the mild NVP group, women in the moderate and severe NVP groups tended to have a greater reduction in dietary food intake (Appendix A).

### 3.4. The Association between NVP and Gestational Weight Gain

Compared with the non-NVP group, women in the severe NVP group had significantly lower weight gain and a slower rate of GWG. The women in the mild and moderate NVP groups, although not statistically different from the non-NVP group, still showed a lower trend. The differences in GWG (*p* = 0.091) and the rate of GWG (*p* = 0.069) between women in the moderate NVP group and the non-NVP group were marginal statistically. Furthermore, the results of Spearman’s analysis showed that with an increase in NVP severity, the GWG of pregnant women was lower, and the GWG rate was slower (Table 6).

## 4. Discussion

This study investigated the association between NVP and dietary intake in first-trimester pregnant women in 10 cities in China. Among the 303 pregnant women enrolled in this study, a total of 84.1% experienced NVP, which is similar to the incidence rates reported in previous NVP studies in the Chinese population: 59.6% [16], 71.8% [12] and 90.9% [17]. Among pregnant women with NVP, the proportions of self-evaluated mild, moderate and severe NVP were 62.4%, 25.9% and 11.8%, respectively. Several other studies of Chinese pregnant women used the Pregnancy-Unique Quantification of Emesis and Nausea (PUQE) scale to assess the severity of NVP. These studies reported a lower proportion of mild NVP (41.4–47.5%) than this study, a higher proportion of moderate NVP (49.2–56.9%) and a similar proportion of severe NVP (1.7–8.7%) [17,18,19], suggesting that the self-assessment by pregnant women tends to be mild compared with the PUQE scale, which may be due to their underestimating the severity of their own NVP.

We found that in the severe NVP group, the rate of GWG was much lower than in the non-NVP group, and half of them even experienced negative GWG. Although our study only examined weight gains in early pregnancy, studies have demonstrated that NVP in the first trimester is associated with lower GWG throughout pregnancy [8,12]. Moreover, studies have shown that insufficient weight gain is associated with a variety of adverse maternal and infant outcomes, such as an increased risk of gestational diabetes [20] and low birthweight or small-for-gestational-age offspring [21,22,23]. Babies born to women with inadequate weight gain had an increased risk of death within their first year of life [24] and slower neurodevelopment later in life [25].

This study found that pregnant women with NVP generally had a lower dietary intake of nutrients, including a variety of macronutrients, vitamins and minerals (i.e., energy, protein, fat, vitamin A, thiamin, riboflavin, vitamin E, phosphorus, potassium, iron, zinc, etc.), compared with those without NVP. We also found that pregnant women with NVP were more likely to have insufficient intake of protein, riboflavin, calcium, phosphorus and selenium. In terms of specific food groups, pregnant women in the NVP group had a lower average daily intake of mushrooms, algae, nuts and seeds, meat, eggs and dairy products. Moreover, the more serious the nausea and vomiting of the pregnant women in the early stage of pregnancy, the lower the intake of energy, all macronutrients and some micronutrients. The intake of nuts, seeds, livestock meat, poultry meat, aquatic products, eggs, dairy products and other food groups also showed a downward trend.

There are great differences in the research conclusions for the impact of NVP on the nutritional intake of pregnant women. Similar to the findings of the present studies, a study conducted in Korea [9] and a study conducted in Finland [10] reported a reduced intake of energy and various nutrients in pregnant women with NVP, including protein, sucrose, vitamin A, thiamin, riboflavin, vitamin B12, vitamin C, niacin, folate, calcium, iron, zinc, etc. Additionally, the more severe the NVP, the greater the reduction in nutritional intake. In terms of specific food groups, the intake of meat products and vegetables in pregnant women with NVP decreased. Furthermore, increased NVP severity was associated with a decreased intake of vegetables, tea/coffee, rice/pasta, breakfast cereals, legumes and citrus fruits/juices and an increased intake of white bread and soft drinks [11]. However, a study conducted in Norway reported significantly different results from the present studies. The energy intake of patients with NVP was higher than that of patients with no NVP, mainly due to the increased intake of carbohydrates and added sugars. After adjustment for energy intake, the NVP group still tended to intake more of almost all types of micronutrients, as well as processed meats, vegetables, fruits and sugar-sweetened soft drinks more frequently [8]. The reason for the inconsistency of results may be that the NVP assessment and nutrient intake assessment methods were different between studies. Moreover, the dietary habits and customs of pregnant women in different regions would also have a significant impact on the research results.

Our study found that NVP patients generally had a lower intake of B vitamins, especially thiamin and riboflavin, which is consistent with other studies. The main food sources of thiamin are grains, beans, nuts, animal offal, lean meat and eggs [26,27]. For pregnant women, the main manifestations of thiamin deficiency are anorexia, nausea and vomiting [27]. Clinically, patients with severe NVP (i.e., HG) have been regarded as a high-risk group for thiamin deficiency [26]. Thiamin is also essential for the fetus, with multiple studies showing that thiamin deficiency has negative effects on fetal brain development and may lead to fetal brain dysfunction [28]. Furthermore, thiamin intake during pregnancy contributes to neuromotor maturation in newborns [29]. Riboflavin is abundant in meat, milk, nuts, eggs and vegetables [30]. Insufficient riboflavin intake during pregnancy may affect embryonic growth and cardiac development in offspring [31]. Furthermore, studies have found that women with postpartum depression had lower serum riboflavin levels [32]. Our study found that the intake of animal meat, nuts and seeds in pregnant women with NVP, especially severe NVP, was lower than in those with no NVP, suggesting that the aversion to these foods may lead to a reduction in thiamin and riboflavin intake, which may have potential adverse effects on the health of both mother and infant.

The potential biological mechanism of the association between NVP with nutritional intake remains uncertain. However, studies have shown that women with NVP have increased olfactory and taste sensitivity [33], which may cause nausea from smells in the environment and from some disgusting tastes in food in order to reject any potentially toxic foods with a strong smell [34]. This statement came from the “maternal–embryo protection hypothesis”, which links NVP to the development of taste aversion. The root of taste aversion is to avoid foods that pose a high threat to the mother and fetus, such as meat, poultry, fish/seafood and eggs [34,35]. Our study also found that the intake of meat and eggs in women with NVP is lower than in those with no NVP. However, these foods tend to be good sources of high-quality protein [36,37] and are essential for fetal growth and development [38]. Therefore, the poor nutrition intake of women with NVP we observed may be due to mothers’ aversion to some certain foods. They choose to avoid or eat certain foods less to control their nausea or vomiting symptoms. However, after all, this study is a cross-sectional study; thus, we cannot deny the possibility that nutrition intake affects NVP symptoms. For example, many studies have proved that vitamin B6 supplements have a good effect on the treatment of NVP symptoms [39]. Similar therapeutic effects were found in quince [40] and ginger [41].

Based on the multicenter data, our study investigated the incidence of NVP in first-trimester Chinese women and explored its association with dietary nutrients and food intake in Chinese women for the first time. The selection of ten cities based on the economic level and geographical location allowed better extrapolation of our study conclusions.

However, this research does have some limitations. Firstly, this study is a cross-sectional study, which limits causal inference. It cannot be excluded that changes in the dietary intake of pregnant women have an impact on the occurrence of nausea and vomiting. Secondly, the survey is mostly based on the self-reported data of pregnant women, which may lead to recall bias. Moreover, an in-person interview may be associated with a higher risk of social desirability bias compared to a self-administered questionnaire. Additionally, nausea and vomiting were not assessed by means of a validated tool such as the PUQE but were self-reported. Finally, the occurrence of maternal nausea or vomiting and dietary intake is a dynamic process, while further prospective cohort monitoring is needed to explore the relationship between them. We also need to explore whether the history of NVP during early pregnancy will have a certain impact on the nutrition intake in the middle or later stage of pregnancy through long-term follow-up observation.

## 5. Conclusions

In conclusion, this study found that the incidence of nausea or vomiting in early pregnancy in China was high. Pregnant women who experienced NVP tend to have a lower intake of multiple foods and nutrients. Additionally, with the increase in the severity of NVP, the decrease in nutrition intake is more serious. Women in the severe NVP group even had insufficient gestational weight gain. This suggests that we need to pay attention to the nutritional intake of pregnant women with NVP and offer targeted nutritional support.

## Figures and Tables

**Table 1 nutrients-15-00933-t001:** Incidence and demographic characteristics of nausea and vomiting during early pregnancy.

	N (%)				*p* †
	Non-NVP (N = 48)	Mild NVP (N = 159)	Moderate NVP (N = 66)	Severe NVP (N = 30)	
Age (years)					0.658
<30	29 (60.4)	98 (62.0)	40 (60.6)	24 (80.0)	
≥30	19 (39.6)	60 (38.0)	26 (39.4)	6 (20.0)	
Education					0.223
Junior high school or below	7 (14.6)	23 (14.5)	6 (9.1)	2 (6.7)	
High school/technical secondary school	5 (10.4)	32 (20.1)	15 (22.7)	7 (23.3)	
College degree or above	36 (75.0)	104 (65.4)	45 (68.2)	21 (70.0)	
Family monthly per capita income (Chinese yuan)					0.925
<5000	16 (34.0)	51 (33.1)	27 (42.2)	7 (24.1)	
5000–9999	24 (51.1)	73 (47.4)	30 (46.9)	17 (58.6)	
≥10,000	7 (14.9)	30 (19.5)	7 (10.9)	5 (17.2)	
Cities					0.031 *
First-tier	5 (10.4)	42 (26.4)	11 (16.7)	3 (10.0)	
New first-tier	22 (15.8)	35 (22.0)	21 (31.8)	10 (33.3)	
Second-tier	12 (25.0)	51 (32.1)	20 (30.3)	11 (36.7)	
Third- or fourth-tier	9 (18.8)	31 (19.5)	14 (21.2)	6 (20.0)	
Pre-pregnancy BMI					0.063
Underweight	4 (8.3)	13 (8.2)	7 (10.6)	2 (6.7)	
Normal	40 (83.3)	107 (67.3)	44 (66.7)	25 (83.3)	
Overweight/obesity	4 (8.3)	39 (24.5)	15 (22.7)	3 (10.0)	
Gravidity					0.204
1	27 (56.3)	73 (45.9)	32 (48.5)	13 (43.3)	
≥2	21 (43.8)	86 (54.1)	34 (51.5)	17 (56.7)	
Parity					0.551
0	31 (64.6)	97 (61.0)	38 (57.6)	18 (60.0)	
≥1	17 (35.4)	62 (39.0)	28 (42.4)	12 (40.0)	

† Chi-square test *p* value. * *p* < 0.05; NVP: nausea and vomiting in pregnancy; BMI: body mass index.

**Table 2 nutrients-15-00933-t002:** Comparison of dietary nutrient intake between pregnant women with and without nausea and vomiting in early pregnancy.

	Median (P^25^, P^75^)		*p* †
	Non-NVP	NVP	
Energy (kcal)	1897.7 (1449.5, 2390.7)	1648.4 (1224.2, 2055.5)	0.009 **
Protein (g)	63.4 (42.0, 86.0)	50.8 (36.8, 69.5)	0.029 *
Fat (g)	77.9 (58.0, 102.5)	55.4 (41.4, 75.2)	<0.001 ***
Carbohydrate (g)	220.6 (167.0, 279.9)	216.2 (149.9, 294.4)	0.400
Dietary fiber (g)	8.6 (6.1, 14.7)	9.1 (5.5, 12.9)	0.512
Cholesterol (mg)	428 (112.4, 768)	312.2 (77.8, 544)	0.078
Vitamin A (μgREA)	399.7 (175.5, 603.9)	266.8 (137.3, 445.4)	0.026 *
Thiamin (mg)	0.8 (0.6, 1.0)	0.7 (0.4, 0.9)	0.006 **
Riboflavin (mg)	0.7 (0.5, 1.2)	0.6 (0.4, 0.8)	0.009 **
Niacin (mg)	10.9 (6.4, 16.3)	9.4 (6.8, 14.3)	0.344
Vitamin C (mg)	58.0 (29.6, 107.1)	53.8 (24.2, 97.5)	0.497
Vitamin E (mg)	31.1 (20.9, 47.8)	22.6 (15.5, 34.7)	0.002 **
Folate (μgDFE)	224.8 (138.8, 347.1)	199.6 (139.6, 303.9)	0.367
Calcium (mg)	375.7 (218.6, 796.7)	308.9 (193.6, 522.4)	0.071
Phosphorus (mg)	816.2 (667.3, 1315.5)	729.4 (538.2, 1016.2)	0.014 **
Potassium (mg)	1816.2 (1314.5, 2975.2)	1517.4 (1083.5, 2108.4)	0.041 *
Sodium (mg)	4132.5 (3532.8, 5237.1)	4001.3 (3071.2, 4893.9)	0.095
Magnesium (mg)	252.4 (178.1, 305.1)	226.6 (166, 308.5)	0.25
Iron (mg)	15.3 (11.4, 20.8)	13 (9.4, 18.1)	0.022 *
Zinc (mg)	8.0 (6.0, 12.9)	6.9 (4.9, 9.7)	0.020 *
Selenium (mg)	33.8 (24.1, 58.3)	31.1 (19.3, 44.2)	0.063
Copper (mg)	1.4 (1.0, 2.3)	1.3 (0.9, 1.9)	0.221
Manganese (mg)	3.1 (2.3, 3.9)	2.7 (1.9, 4.0)	0.278

† Mann–Whitney U test *p* value. * *p* < 0.05; ** *p* < 0.01; *** *p* < 0.001. NVP: nausea and vomiting in pregnancy.

**Table 3 nutrients-15-00933-t003:** Comparison of dietary nutrient intake between the NVP and non-NVP groups after adjusting for demographic characteristics.

	Adjusted β Value (95% CI)		Wald Value	*p* †
	Non-NVP	NVP		
Energy (kcal)	[Reference]	−299.3 (−510.8, −87.7)	7.688	0.006 **
Protein (g)	[Reference]	−18.6 (−29.1, −8.1)	11.985	0.001 **
Fat (g)	[Reference]	−16.4 (−25.7, −7.1)	11.994	0.001 **
Carbohydrate (g)	[Reference]	−25.6 (−57.7, 6.4)	2.454	0.117
Dietary fiber (g)	[Reference]	−2.6 (−5.2, 0.1)	3.489	0.062
Cholesterol (mg)	[Reference]	−87.6 (−197.4, 22.2)	2.444	0.118
Vitamin A (μgREA)	[Reference]	−95.3 (−356, 165.3)	0.514	0.473
Thiamin (mg)	[Reference]	−0.2 (−0.4, −0.1)	7.577	0.006 **
Riboflavin (mg)	[Reference]	−0.3 (−0.5, −0.2)	13.355	<0.001 ***
Niacin (mg)	[Reference]	−1.9 (−4.5, 0.7)	2.090	0.148
Vitamin C (mg)	[Reference]	4.6 (−32.3, 41.5)	0.060	0.807
Vitamin E (mg)	[Reference]	−7.2 (−12.8, −1.6)	6.312	0.012 *
Folate (μgDFE)	[Reference]	−36.8 (−89.0, 15.3)	1.914	0.166
Calcium (mg)	[Reference]	−200.8 (−327.2, −74.5)	9.708	0.002 **
Phosphorus (mg)	[Reference]	−256.8 (−392.8, −120.8)	13.702	<0.001 ***
Potassium (mg)	[Reference]	−462.7 (−843.7, −81.7)	5.665	0.017 *
Sodium (mg)	[Reference]	−235.7 (−803.7, 332.3)	0.661	0.416
Magnesium (mg)	[Reference]	−40.5 (−88.6, 7.5)	2.733	0.098
Iron (mg)	[Reference]	−6.7 (−11.5, −2.0)	7.634	0.006 **
Zinc (mg)	[Reference]	−2.2 (−3.5, −0.8)	10.264	0.001 **
Selenium (mg)	[Reference]	−14.9 (−28.3, −1.5)	4.750	0.029 *
Copper (mg)	[Reference]	−0.3 (−0.9, 0.4)	0.781	0.377
Manganese (mg)	[Reference]	−0.6 (−2.4, 1.3)	0.367	0.545

† Multiple linear regression *p* values after adjustment for age, education, income, city, pre-pregnancy BMI, gravidity and parity. * *p* < 0.05; ** *p* < 0.01; *** *p* < 0.001. NVP: nausea and vomiting in pregnancy.

**Table 4 nutrients-15-00933-t004:** Comparison of dietary nutrients’ adequacy rate between pregnant women with and without nausea and vomiting in early pregnancy.

	N (%)		*p* ^a^	Adjusted OR (95% CI) ^b^	Adjusted *p*
	Non-NVP	NVP			
Protein	27 (56.3)	106 (41.6)	0.060	0.45 (0.22, 0.90)	0.023 *
Vitamin A	8 (16.7)	29 (11.4)	0.304	0.73 (0.28, 1.90)	0.518
Thiamin	11 (22.9)	32 (12.5)	0.059	0.49 (0.21, 1.15)	0.099
Riboflavin	11 (22.9)	24 (9.4)	0.007 **	0.34 (0.14, 0.83)	0.018 *
Niacin	21 (43.8)	92 (36.1)	0.313	0.60 (0.30, 1.19)	0.142
Vitamin C	13 (27.1)	60 (23.5)	0.597	0.89 (0.41, 1.95)	0.778
Vitamin E	44 (91.7)	208 (81.6)	0.086	0.47 (0.16, 1.43)	0.184
Calcium	12 (25.0)	24 (9.4)	0.002 **	0.30 (0.12, 0.71)	0.007 **
Phosphorus	34 (70.8)	133 (52.2)	0.017 *	0.32 (0.15, 0.69)	0.004 **
Potassium	21 (43.7)	77 (30.2)	0.066	0.53 (0.27, 1.07)	0.075
Magnesium	10 (20.8)	42 (16.5)	0.462	0.63 (0.27, 1.50)	0.301
Iron	13 (27.1)	46 (18.0)	0.147	0.53 (0.24, 1.16)	0.110
Zinc	17 (35.4)	65 (25.5)	0.156	0.50 (0.24, 1.04)	0.063
Selenium	11 (22.9)	29 (11.4)	0.030 *	0.39 (0.16, 0.96)	0.041 *
Copper	37 (77.1)	192 (75.3)	0.791	0.96 (0.44, 2.12)	0.927
Manganese	7 (14.6)	37 (14.5)	0.989	1.02 (0.38, 2.71)	0.971

^a^ Chi-square test *p* value. ^b^ Using the logistic regression model whose reference category is non-NVP. Adjustment factors include age, education, income, city, pre-pregnancy BMI, gravidity and parity. * *p* < 0.05; ** *p* < 0.01; NVP: nausea and vomiting in pregnancy; OR: odds ratio.

**Table 5 nutrients-15-00933-t005:** Comparison of average daily food intake of pregnant women with and without nausea/vomiting in early pregnancy (g/d).

	Median (P^25^, P^75^)		*p* †
	Non-NVP	NVP	
Cereals	198.0 (120.4, 289)	205.0 (113.6, 320.8)	0.649
Potatoes	32.7 (14.0, 70.0)	23.3 (6.7, 55.3)	0.082
Fresh vegetables	160.0 (58.3, 300.0)	120.0 (55.0, 240.0)	0.241
Pickled and fermented vegetables	0.3 (0.0, 4.7)	0.2 (0.0, 3.5)	0.346
mushroom and algae	11.7 (4.1, 20.1)	6.7 (1.4, 18.5)	0.039 *
Soybeans and their products	20.0 (8.2, 48.0)	17.7 (5.8, 40.2)	0.409
Nuts and seeds	22.7 (7.9, 40.0)	10.5 (1.3, 25.0)	0.001 **
Fruits	300.0 (200.0, 500.0)	250.0 (150.0, 426.0)	0.188
Poultry meat	38.5 (14.2, 97.5)	20.0 (2.0, 50.0)	0.003 **
Livestock meat	11.0 (3.3, 43.8)	7.0 (0.0, 23.3)	0.063
Aquatic products	14.6 (3.8, 68.8)	14.0 (2.7, 30.3)	0.202
Eggs	60.0 (46.4, 60.0)	49.0 (17.5, 60.0)	0.007 **
Dairy products	250.0 (101.3, 295.4)	159.7 (30.0, 261.3)	0.032 *
Soups	44.2 (0.0, 186.7)	40 (0.0, 116.7)	0.336
Snacks and beverages	5.0 (0.0, 46.8)	2.2 (0.0, 22.6)	0.213

† Mann–Whitney U test *p* value. * *p* < 0.05; ** *p* < 0.01; NVP: nausea and vomiting in pregnancy.

**Table 6 nutrients-15-00933-t006:** Comparison of gestational weight gain with severity of nausea and vomiting.

	Median [P^25^, P^75^]				Spearman’s *p*
	Non-NVP	Mild NVP	Moderate NVP	Severe NVP	
GWG (kg)	1.00 [0.00, 2.95]	1.40 [0.00, 3.00]	0.50 [−0.24, 1.83]	0.00 [−1.15, 1.63] †	0.003 **
rGWG (kg/week)	0.13 [0.00, 0.32]	0.12 [0.00, 0.27]	0.04 [−0.02, 0.16]	0.00 [−0.12, 0.15] †	0.001 **

† Compared to the non-NVP group: Mann–Whitney U test *p* < 0.05. ** *p* < 0.01; NVP: nausea and vomiting in pregnancy; GWG: gestational weight gain; rGWG: the rate of GWG per week.

## Data Availability

The data presented in this study are available on request from the corresponding author. The data are not publicly available due to ethical restrictions.

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
