# Peer review of "Nausea and Vomiting during Early Pregnancy among Chinese Women and Its Association with Nutritional Intakes"

_nutrients, 2023, doi:10.3390/nu15040933_

Round 1
Reviewer 1 Report
Major revisions
The article deals with the interesting issue of nutritional intake in relation to the occurrence of different degrees of early gestosis. In my opinion, the article does not have sufficient scientific potential, i.e., it does not bring knowledge that would significantly enlighten the audience. Overall, it is described those women with NVP had lower nutrient intakes. In addition to the description of the intake of individual nutrients and foods in a selected population of women, which are beneficial, it is probably not surprising that pregnant women suffering from nausea and vomiting have a lower appetite and the resulting lower intake of nutrients.
It would be significantly more beneficial if the article evaluated the nutritional intake during the entire pregnancy, where it would be seen if the nutrient intake increases after the end of early gestation and if the body will compensate for the losses from the first trimester. At the same time, it would be interesting to analyze nutritionists from the given country and individual regions.
Minor revisions
The article is not prepared carefully:
· in 3.1, the specification of the criteria with the citation, according to which the women were classified into different degrees of NVP, is missing,
· table 1 (error in sum 98+60 = 158, not 159)
· the tables lack a legend with explanations of abbreviations (each table must be comprehensible in terms of content even outside the text),
· in tab. 2 and 3 missing units (kcal, g).
· It is appropriate to supplement and develop multiple linear regression and its mathematical expression (this way it is not completely transparent),
· table 5 (care in parentheses)
Author Response
评论1:本文涉及与不同程度的早期妊娠症发生有关的营养摄入的有趣问题。在我看来,这篇文章没有足够的科学潜力,即它没有带来可以显着启发观众的知识。总体而言,描述那些患有NVP的女性营养摄入量较低。除了描述特定女性人群中个别营养素和食物的摄入量是有益的之外,患有恶心和呕吐的孕妇食欲较低,从而导致营养摄入减少,这可能并不奇怪。
回应1:感谢您的宝贵反馈,非常感谢。正如你所指出的,患有恶心和呕吐的孕妇食欲较低,从而导致营养摄入减少,这可能并不奇怪。但是我们的研究可以探索哪些营养素和食物的减少至关重要。基于这一结论,为患有NVP的孕妇提供一些有针对性的营养支持,例如提供特定的营养补充剂或食品替代品。我们的研究包括中国的十个城市,这些城市可以很好地代表中国早期孕妇的情况。我们的研究首次探讨了NVP对中国人群营养摄入的影响。由于不同国家独特的饮食习惯,在中国人中进行调查是有意义的。通过对NVP孕妇膳食摄入的初步了解,观众可以更好地为孕期营养支持提供有针对性的建议。
评论2:如果文章评估了整个怀孕期间的营养摄入量,那么将更加有益,在那里可以看到早期妊娠结束后营养摄入量是否增加,以及身体是否会补偿孕早期的损失。同时,分析来自特定国家和个别地区的营养学家会很有趣。
回应2:感谢您的善意建议。然而,这项研究是一项横断面研究,不可能评估整个怀孕期间的营养摄入量。这一点在文章的限制部分(第9页第374至379行)中也提到。我们还准备进行一项队列研究,以进行长期随访观察。您的建议将对我们的研究有很大帮助。
评论3:在3.1中,缺少引用标准的具体说明,根据该标准,妇女被分为不同程度的NVP。
回应3:非常感谢您的建议。根据您的意见,我们添加了 NVP 严重性标准(第 144 页的第 145 行至第 3 行)。此外,我们还在材料和方法部分(第 3 页第 103 至 104 行)解释了分类标准。
注释4:表1(总和误差98+60 = 158,而不是159)
回应4:感谢您的评论。我很抱歉,但也许我没有说清楚。表1中98+60≠159的原因是轻度NVP组孕妇的年龄数据缺失。
评论5:表格缺少带有缩写解释的图例(即使在文本之外,每个表格也必须在内容上易于理解),
回应5:谢谢你指出这一点。我们已经在每个表格的注释部分解释了缩写。
注释6:在选项卡2和3中缺少单位(千卡,g)。
回应6:谢谢你的建议。我们在修订版的表格中添加了营养素的单位。
注释7:补充和发展多元线性回归及其数学表达式是适当的(这样它就不完全透明了),
回应7:非常感谢您的建议。然而,由于每种营养素都有自己的多元线性回归模型,因此考虑到长度,它们的数学表达式无法完全呈现。因此,我们在修改后的版本中临时添加了Wald c2以供参考(表3)。如果您认为我们仍然需要一个完整的多元线性回归模型,我们可以在附录中为您提供。希望能得到您的理解。
Comment 8: table 5 (care in parentheses)
Response 8: Thank you for pointing out this point that needs attention. We have made corresponding modifications according to your suggestions (in the new version of Table 6).
We sincerely hope that this revised manuscript has addressed all your comments and suggestions. We appreciated for your warm work earnestly, and hope that the correction will meet with approval. Once again, thank you very much for your comments and suggestions.

Reviewer 2 Report
Thank you for the opportunity to revise this interesting paper.
One of the main criticism is the high rate of missing data. Actually, 2/3 of the sample was lost because there was no data on nausea and vomiting. It appears to be strange considering the aim of the study. How could you explain this aspect? It could highly impact the representativeness of the sample. I suggest showing descriptive characteristics of participants with and without missing data in a supplementary table. Please, also add data interpretation, if any differences between the two groups. In addition, please address it in the limitations section of the manuscript.
Could you also explain why there is a so high missing data rate considering that data collection was performed through a face-to-face interview? It is really strange.
lines 90-94: "With the aid of a standard food reference map, standard bowls and plates, a one-time 24-hour dietary recall was used to record the types and quantities of all foods, beverages and condiments consumed by the participants over the previous 24 hours. " Please add references for the tools used. if they are not published, please add a brief description and consider to add as supplementary materials/appendix. The same also for the FFQ used.
Food categorization should be detailed. as for instance, what did you include in cereals? and what about fresh vegetables, and so on?
Line 103: "The trained investigators used standard tools to measure and record the height and weight of pregnant women." As previously required, specify which tools with reference and brief description.
Statistical analysis: no information regarding sample size calculation is reported. please add.
A "Potential biological mechanisms" that explain the observed associations should be added. Actually, in the current version of the discussion, the authors reported references assessing the association between nutrient intake and maternal-fetal health outcomes, but not in relation to nausea and vomiting. Please add.
In the conclusion, the sentence: "In conclusion, this study found that the incidence of nausea or vomiting in early pregnancy in China was high" should be revised. Actually, as mentioned before 2/3 of the initial sample has been removed from the analysis because of missing information on the outcome. It means that your data should be interpreted with more caution.
A limitations section should be added. Actually, several aspects limit the generalizability of your results. First, it is a cross-sectional study. Second, previous studies found that an in-person interview is associated with a higher risk of social desirability bias compared to a self-administered questionnaire. Moreover, nausea and vomiting were not assessed by means of a validated tool (despite availability) but were self-reported. This could be another aspect linked with the high rate of nausea and vomiting observed.
In the conclusions, the sentence: "The intake of dietary nutrients and food groups by pregnant women with NVP decreased in varying degrees." need more clarification.
Author Response
Comment 1: One of the main criticism is the high rate of missing data. Actually, 2/3 of the sample was lost because there was no data on nausea and vomiting. It appears to be strange considering the aim of the study. How could you explain this aspect? It could highly impact the representativeness of the sample. I suggest showing descriptive characteristics of participants with and without missing data in a supplementary table. Please, also add data interpretation, if any differences between the two groups. In addition, please address it in the limitations section of the manuscript.Could you also explain why there is a so high missing data rate considering that data collection was performed through a face-to-face interview? It is really strange.
Response 1: Thank you for this valuable feedback, which is highly appreciated. Firstly, we are very sorry. Maybe it's because we didn't express it clearly. Our research is aimed at women in the first trimester of pregnancy. In the Subjects section, the 934 respondents we mentioned are women in the first, second and third trimester of pregnancy. For the purpose of our study, we only selected women in the first trimester of pregnancy for analysis. It is not because 2/3 of the samples are missing NVP data. Actually, of 307 women in the first trimester of pregnancy, only 4 women had no NVP data. In the revised manuscript, we changed the way of expression to avoid possible misunderstanding (Line 72 to 79 in Page 2).
Comment 2: lines 90-94: "With the aid of a standard food reference map, standard bowls and plates, a one-time 24-hour dietary recall was used to record the types and quantities of all foods, beverages and condiments consumed by the participants over the previous 24 hours. " Please add references for the tools used. if they are not published, please add a brief description and consider to add as supplementary materials/appendix. The same also for the FFQ used.
Response 2: Thank you for your suggestion. For the tools we use, we have given the corresponding references. (Line 106 in Page 2).
Comment 3: Food categorization should be detailed. as for instance, what did you include in cereals? and what about fresh vegetables, and so on?
Response 3: Thank you for pointing out this point. The specific classification method has been described in Table S1 in the appendix.
Comment 4: Line 103: "The trained investigators used standard tools to measure and record the height and weight of pregnant women." As previously required, specify which tools with reference and brief description.
Response 4: Thank you for your suggestion. We have briefly described the tools used in the modified version (Line 119 to 120 in Page 3).
Comment 5: Statistical analysis: no information regarding sample size calculation is reported. please add.
Response 5: Thank you for your valuable comments. According to your opinion, we have added the sample size calculation part (Line 81 to 88 in Page 2).
Comment 6: A "Potential biological mechanisms" that explain the observed associations should be added. Actually, in the current version of the discussion, the authors reported references assessing the association between nutrient intake and maternal-fetal health outcomes, but not in relation to nausea and vomiting. Please add.
Response 6: Based on your valuable comments, we have added an explanation of the potential biological mechanism in the revised discussion section (Line 331 to 347 in Page 9).
Comment 7: In the conclusion, the sentence: "In conclusion, this study found that the incidence of nausea or vomiting in early pregnancy in China was high" should be revised. Actually, as mentioned before 2/3 of the initial sample has been removed from the analysis because of missing information on the outcome. It means that your data should be interpreted with more caution.
Response 7: Thank you for pointing this out. We hope that our explanation in Response 1 can answer this question.
Comment 8: A limitations section should be added. Actually, several aspects limit the generalizability of your results. First, it is a cross-sectional study. Second, previous studies found that an in-person interview is associated with a higher risk of social desirability bias compared to a self-administered questionnaire. Moreover, nausea and vomiting were not assessed by means of a validated tool (despite availability) but were self-reported. This could be another aspect linked with the high rate of nausea and vomiting observed.
Response 8: Thank you for your comments. We added the limitations section in the revised version (Line 435 to 446 in Page 9).
Comment 9: In the conclusions, the sentence: "The intake of dietary nutrients and food groups by pregnant women with NVP decreased in varying degrees." need more clarification.
答复9:我们非常抱歉,我们的声明不是很清楚。根据您的意见,我们进行了修改:“经历过NVP的孕妇往往对多种食物和营养素的摄入量较低,并且随着NVP严重程度的增加,营养摄入的减少更为严重”(第9页第450至452行)。
我们衷心希望这份修订稿能解决大家的所有意见和建议。我们衷心感谢您的热情工作,并希望更正得到认可。再次非常感谢您的意见和建议。

Round 2
Reviewer 1 Report
Although I have a different opinion on some aspects of the article, the additions and corrections have made the results of the described study more accurate and more clearly describe the conclusions drawn from the study. From the above, I agree with its publication.
Author Response
Thank you for agreeing to publish this article. We sincerely appreciate your kind suggestions and comments.
Reviewer 2 Report
Line 82, for more clarity I would suggest substituting samples with women or participants.
Line 88 typing error
Author Response
Point 1: Line 82, for more clarity I would suggest substituting samples with women or participants.
Response 1: Thank you for your valuable comments. But maybe our word version is different. In the version I downloaded, it may be Line 80 that need to be replaced. According to your opinion, we have substituted samples with participants.
Point 2: Line 88 typing error
Response 2: Thank you for pointing this out. We have corrected the typing error in Line 88.
We sincerely hope that this revised manuscript has addressed all your comments. Once again, thank you very much for your warm comments and suggestions.
